

**"Unless someone like you cares a whole awful lot, nothing is going to get better":**
**An environmental discourse analysis of animated films *The Lorax* (2012) and *Tomorrow***
**(2019)**
*Mohammad Mizan-Rahman*
*PhD Candidate, American Culture Studies*
*School of Cultural and Critical Studies, Bowling Green State University*
*mohrah@bgsu.edu*
**Abstract**
Using environmental humanities discourse analysis, this article asks how environmental
issues are exhibited in two environmentally focused animated films, *The Lorax* and
*Tomorrow*, produced in Hollywood (United States) and Dhallywood (Bangladesh),
respectively, and how people responded to these films on social media websites. The first part
of the article is the analysis of selected social media pages to understand the impact of these
two films on contemporary environmental discourse, and the second part comprises an
analysis of the environmental narrative of the films. I selected these two films for four
reasons: i) they are both environmental educational and pedagogical tools; ii) they use
environmental storytelling; iii) they both address sustainability; and iv) they may have
influenced some discourse on environmental issues on social media. The study demonstrates
that environmentally driven animated films can shape the discourse of their audiences. This
study also demonstrates how narratives from films such as *The Lorax* and *Tomorrow* can lead
an audience to consider large-scale environmental issues.
**Keywords**
environmental storytelling; environmental discourse analysis; environmental education;
environmental communication; sustainability; sense of place; films; social media



## 1. Introduction

In *Sense of Place, Sense of Planet* (2008), environment and sustainability scholar
Ursula Heise demonstrates how the public can be influenced by environmental narratives.
Because artwork can both reflect or influence cultural events and trends, they are a useful tool
to understand people's views and opinions about the environment specifically by examining
how they shape public perception by developing and promoting certain kinds of
environmental discourse.
Using environmental discourse analysis as a narrative inquiry, I investigated two
animated eco-blockbusters: *The Lorax* (Chris Renaud, 2012) and *Tomorrow* (Mohammad
Shihab Uddin, 2019).[1] The films were produced in Hollywood (United States) and
Dhallywood (Bangladesh, Bangladeshi production house Cycore Studios), respectively.
Animated films may be a powerful medium of environmental education and shape the public
discourse, as discussed below. While *The Lorax* describes the severity of waste and
environmental collapse caused by deforestation (and implicitly climate change, given the
media environment and promotional material around the release of the film),
*Tomorrow* describes how such events result in climate change. *The Lorax* uses a fictional
world to deliver a general message while *Tomorrow* highlights the reality of severe climate
injustices in the global South such as Bangladesh.
Humanities scholars such as Alexander Elliott and James Cullis (2017) argue that
research on climate change has shifted to a global scale from a previous focus on the Euro-
American perspective. The film *Tomorrow* reflects this trend in the realm of popular culture.
*Tomorrow* came out in 2019, after Bangladesh had experienced several environmental
disasters, including flash floods. 2012's *The Lorax* is similar, despite not being set in a

---

[1]Although the television version of *The Lorax* produced by DePatie-Freleng Enterprises is closer to *Tomorrow* in terms of time format, the 1972 TV version is not suitable for a social media analysis due to the time of its release risking viewer nostalgia being a factor in the discourse, potentially compromising the analysis.



specific place, in that it was released at a time when environmental catastrophes including
earthquakes, wildfires, and hurricanes were major stories in media across the globe, including
the previous year's Fukushima disaster. Both films therefore addressed the global nature of
environmental crisis in a timely manner. Through the joint analysis of the films and their
reception by viewers on social media, this study finds evidence that these two films gave their
viewers thematic narratives and talking points that they then incorporate into personal
discussion and in general promotion of environmental causes.

**2. Methodology**

Using environmental humanities discourse analysis as a tool, the principal question of

this study is: *How do* The Lorax *and* Tomorrow *instruct viewers about key environmental*
*messages?* To answer this overarching question, I consolidated the public comments and the
narrative analysis of the films into three main categories: *environmental catastrophe,*
*environmental storytelling,* and *environmental education* in order to address three questions
related to these three environmental discourses. First, *how are these two films situated within*
*the discourses of environmental catastrophe*? Second, *how do these two films perform*
*environmental storytelling while emphasizing a sense of place*? Third, *what sorts of*
*educational messages do these two films spread regarding sustainability*? I use an
environmental discourse analysis model drawn mainly from anthropologists Peter
Mühlhäusler's and Adrian Peace's scholarship. My narrative analysis also incorporates
spatiality as it shows how discourse may vary in different local, regional, and global contexts
even when they address the same environmental concerns. While discussing the methods of
environmental discourse analysis, Peace states that emphasis on keywords and select
terminologies is vital to the anthropological contribution to environmental discourse analysis
(p. 415). As a part of the environmental discourse analysis, I chose the selected words and



phrases from the content and comments of the two films. The study also deals with spatiality
as both comments and contents highlight local and global concerns about the environment.
The environmental discourse analysis in the paper is structured as follows. First, this
paper provides a brief synopsis of the films. Second, this study considers the literature on
how public comments online pertain to broader environmental media. Continuing the
discourse analysis, a select sampling of activity on social media pages related to each film is
analyzed to understand the discourse surrounding each. Third, the paper provides an
environmental discourse analysis to extract the themes and narratives from both films with
the most impact. By using both an analysis of social media posts about the films and a direct
analysis of the films themselves, this study demonstrates how aspects of each film influenced
public discourse.
I use social media as a platform to measure and understand public reactions. To
extract public comments about *The Lorax* on social media, I used the search term: "lorax" on
Twitter using the Netlytic social networks analyzer, which yielded exactly 1000 comments.
This number of comments was chosen as it is the default used by Netlytic and represents a
reasonable sample for manual coding of sentiments. I confined the study to Twitter because
Netlytic does not extract comments from Facebook, and because *The Lorax* Facebook page
has very few public comments from which to glean data. Furthermore, *The Lorax* does not
have a YouTube page. For *Tomorrow,* I extracted comments from *Tomorrow's* YouTube
page, as *Tomorrow* does not have either Facebook or Twitter pages. I extracted comments by
using a web scraping method written in the Python programming language, using the search
term: "tomorrow animated movie"; which yielded 1510 comments (Bengali and English) out
of 4974 total, based on which comments received more "likes" (the remainder of comments
were omitted for falling below a threshold of likes). For *The Lorax*, the Twitter comments
spanned a decade, as the film was released in 2012. As *Tomorrow* is a relatively recent





release, so are all its comments. After transferring the data to a Microsoft Excel spreadsheet, I
manually examined it, developing codes to analyze public reactions pertaining to different
environmental discourses. I began with a total of twenty-five codes, re-examined the data,
and condensed them to seven major codes. Using the statistical programming language R, I
present a graphical text categorization algorithm that generates skip-gram phrases selectively,
by extracting and using phrases. Commenter names and online handles have been excluded
for anonymity.

**3. Film synopses**

The animated film *The Lorax* (2012) is loosely based on Dr. Seuss's children's book,

*The Lorax* (1971), although the plot of the film diverges from the source material. Visual
communication studies scholar Dylan Wolfe (2008) notes that environmentalism is a key
feature of the work.

Produced by Illumination Entertainment and released by Universal Pictures in 2012,

*The Lorax* had a budget of $70 million and grossed $348.8 million worldwide. It showcases
the process of industrialization, portraying the cause and effect of the hypocritical nature of
"human progress" when externalities are not addressed, and the environment is not thought of
as worth protecting. It must be noted that environmental messaging in media adjacent to the
film was compromised, as noted by Caraway and Caraway (2020) in their article, while the
film railed against greenwashing, cross promotion in advertising with the film was used to
showcase gas-powered automobiles. In her article, communication scholar Ellen Moore
(2016) also notes this as a flaw as it changes the focus from reducing consumption to
encouraging a nebulous "green" consumption.

Ted, the protagonist, wants a real tree, which is now so rare as to be mythical. The

tree is intended to impress a girl he likes named Audrey. Audrey personifies trees and is





described using words such as "softer than silk" and "smelled like butterfly." Because of

Audrey's love of nature, her character invokes a sense of conservation. Audrey shows Ted a

painting of trees with a sense of loss and lamentation. Unlike how the general population and

especially the industrialists of their town of Thneedville approach trees, Audrey's approach

produces a renewal in environmental consciousness. To investigate the disappearance of the

real trees, Ted visits a hermit known as the Once-ler and the Lorax who "speaks for the

trees." Speaking for the trees (Earth) is portrayed in a positive light.

  The Once-ler represents industrialist society, which profited from development, but at

the cost of pollution and deforestation. The Once-ler employs subterfuge in his

industrialization, including "greenwashing." The Lorax's warnings were ignored by the

Once-ler when he became an industrialist, and the sky was filled up with smoke, the water

polluted with sludge, and the land was left barren. Greed and the illusion of progress

deafened the Once-ler to the words of the Lorax until one day the last Truffula tree was

chopped down, and the Once-ler discovers that he is condemned to grow old and waste away

in the wretched badlands of his own making. This very clear cause and effect is a cautionary

tale to viewers, showing how unethical profiteering can one day leave them worse off, with

gains that were fleeting. Because the Lorax disappeared when the last Truffula tree was

chopped down, the Once-ler relays to Ted the Lorax's cryptic last message, "Unless someone

like you cares a whole awful lot, nothing is going to get better, it's not" (1:02:09). This is a

clear call to action to the audience, as Ted is the archetype of the everyman, a person who the

audience can relate to. Indeed, the Once-ler charges Ted, and by extension the audience, with

repairing the devastated environment. However, other industrialists in the movie, chief

among them a clean air tycoon named O'Hare, fight to keep the status quo by tricking the

populace, subconsciously warning the audience that pushback from people they know may in

fact be misdirection from real-life industrialists.



The 2019 film *Tomorrow* (budget: 10 million BDT, converted roughly to ~119,000
USD; a reliable figure for gross income could not be found) similarly portrays a dire future in
the hopes that the present generation will find a way to avoid it. *Tomorrow* begins when
Ratul and his father, a nature lover, learn that sea levels are rising, which will make them and
their fellow villagers ecological refugees. Despite this knowledge, they and the villagers are
reluctant to take any actions to prevent the hazard. One of the villagers' comments, "why
would we ruin today thinking about tomorrow?" (4:10) is a refrain viewers may be familiar
with, a carelessness about their own future.
In a dream, Ratul learns that Bangladesh is going to face a disastrous fate because of
rising sea levels combined with the melting ice caps in the Himalayas. To answer Ratul's
questions, Batasher Buro, a shamanic figure known as "the Old Man of the Winds," takes
him to the future, in which most of southern Bangladesh is submerged, with almost 30
million homeless and destitute people taking shelter in the north. This reflects real life—for
some time now, residents of southern Bangladesh have been migrating to the capital city,
Dhaka, and other comparatively highland parts of the country. But there is still hope: the Old
Man of the Winds takes Ratul to another possible future, where solar panels and windmills
are commonplace and there is no usage of fossil fuels. This alternate future implies that
mankind has a choice. Ratul wants to know how to build a future like this, but the Old Man
of the Winds leaves, saying this is Ratul's planet and he himself needs to seek an answer.
Similar to *The Lorax*, Ratul is an everyman, with the audience implicitly being told to
personally care about the environment themselves.
Ratul awakens concerned about the welfare of Earth. Inspired by his father's
motivational speech about saving the environment, Ratul starts a campaign on social media
focused on taxing fossil fuels, inspiring protests, which start taking place all over the world.
(Posting on social media and protesting are actions that viewers may be able to take on their



own; these easier actions are shown first, lowering the barrier for meaningful action by the
audience.) The film then leaps 25 years in the future to show a grownup Ratul delivering a
speech at the United Nations. By then, many parts of the world, including southern
Bangladesh, are submerged. But there is optimism that Bangladesh can rehabilitate its people
with money from a tax on fossil fuels; the other countries of the United Nations begin helping
to address the climate crisis, following in the footsteps of Bangladesh. Ratul hears the voice
of the Old Man of the Winds, who tells him that he has been successful in saving the world.
This is more than a narrative statement; it is a clear statement to the audience that their
actions have the potential make a real impact.

**4. Analysis of public reactions to the films on Twitter and YouTube**

The audience is a key part of the environmental discourse equation, and these two

films generated many positive reviews on the social media pages related to the films.
Audiences' reactions to media are important to understand so that artists, activists, and
academics may even more effectively contribute to environmental awareness. Despite this
clear need, some scholars caution that we lack sufficient knowledge regarding how audiences
react to environmental communication, calling for more such studies (Kluwick, 2014;
Garrard, 2014, p. 20). Solitary public comments on social media may be inconsequential on
their own, but together, they are important to understand public reception. Unlike formal
media, informal social media is often free from the traditional trappings of media criticism;
the opinions on social media are often that of laypeople who are concerned with different
aspects of the film than a professional critic would be. Furthermore, the opinion of a friend or
family member on social media may have more impact on someone than that of a distant
critic whom the reader does not know. Social media comments are not a perfect stand in for
an "average" opinion of the film, as social media posts come with their own biases, and there





200 are economic and geopolitical factors that affect who is able to access the internet, and by

201 extension social media platforms.

202  I manually examined each of the selected 1510 public comments about *Tomorrow*

203 (beginning with the comment with the most "likes" (1.4K), ending with those with just one).

204 The most-liked comment states that *Tomorrow* is a locally made film with a global

205 international standard that carries an environmental narrative. The most-liked comments after

206 that are about the quality of the film and that it deserves international accolades. The major

207 seven environmental discourses derived from such public comments are presented in Table 1.

208    **Table 1: Example of YouTube Comments from *Tomorrow***

209     ***(Arranged by total number of likes in sample)***

| Discourse | Total number of likes of the combined comments | Total number of comments | Example of comments[2] |
|---|---|---|---|
| Environmental education | 1541 | 131 | "It should be premiere in every School in Bangladesh … It's the most Realistic animated short movie I ever seen!" |
| Climate change | 1239 | 95 | "This film … [shows] examples of how climate change can affect us environmentally and as a community" |
| Sense of place | 142 | 116 | "Local places are getting destroyed because of global places" (my translation) |
| Environmental activism | 191 | 88 | "It's our duty to save our world, to save our people to save the wildlife # stand Against fossil fuel 👊 # Raise awareness among all the people✋" |
| Environmental storytelling | 189 | 309 | "Story is beautiful…I love this story…" |
| Sustainability | 152 | 158 | "If we plants tree more, one day we get a beautiful Bangladesh. Let's go we plants tree for a beautiful future" |
| Plastic/waste | 110 | 29 | "We should not destroy the environment by producing plastic" (my translation) |


---

[2] Except for my translations, grammatical and spelling errors in comments have been left as is.



Public reactions juxtapose positive reviews of the content of the film with negative
statements about the current local and global environmental conditions. Online commenters
urge showing *Tomorrow* in all primary and secondary schools across Bangladesh while
encouraging elected officials to watch and screen it as well. This commentary connects with
the film narrative as the protagonist of *Tomorrow* is a schoolchild, who goes on an abridged
hero's journey to affect global politics regarding environmental laws and policies.
The comments for *Tomorrow* demonstrate the power of locally produced media, an
aspect of the film which may prove valuable to other environmental communicators and
educators. Climate communication scholars Candice Howarth and Alison Anderson (2019)
highlighted that stronger collaborative bonds between local media and scientific research
helps form a more trusted relationship between local media and other local stakeholders and
increases engagement with climate change. Many comments express an emotional response
to seeing environmental destruction in their own localized area, for example, "Alas! My
home is in Hatia, the southern part of Bangladesh" (my translation). Since *Tomorrow* was
made in Bangladesh, it may create a greater local impact than if it had been produced in
Hollywood, or even Bollywood. Relatedly, Howarth and Anderson (2019) have noted that
climate change is often understood as "abstract and distant" (p. 718). *Tomorrow*, by contrast,
shows how climate change is an issue requiring both local and global action.
Earth's restoration is possible only if we can imagine it clearly. Stories occupy an
important role in that ability. For all of us, stories matter; if we know our local story[3],
especially through local media, we can participate in a range of actions to restore our local
landscapes. However, the public of Bangladesh often do not believe that their elected
officials will reduce the use of fossil fuels on their own. Yet the movie instills civic hope in
some viewers—comments like "should the Prime Minister watch this movie, the country

---

[3] That is to say, how our local environment came to be, and how our actions alter it.



would benefit greatly" (my translation) reinforce the position in the public discourse that the
ordinary people of Bangladesh do not trust the government to take adequate action.
Intriguingly, this mirrors the events of the film, where the government increases taxes on
fossil fuels following public outcry—showing that such a strategy is viable in the real world.

Public comments carry a sense of responsibility and an ethics of care. Commenters

use words and phrases such as "I cannot control my emotions," "my eyes were moistened
while watching the movies," "the Earth needs to be protected," "save Earth, save
Bangladesh" and many more, explain their sense of empathy and responsibility with local and
global places. These are virtual comments, yet this sense of awareness is exhibited,
nonetheless. This is again demonstrated when commenters give attention to the occasional
hypocrisy or at least inconsistencies in the story. They appreciate the protest against fossil
fuels, and critique Ratul's flight on a fossil fuel–powered airplane; they appreciate the
message about the environment.

Regarding environmental activism, there were many comments such as "we each have

a responsibility to save the world," and "all mass media ought to disseminate this film
massively to create public awareness regarding climate change… the UN must force a new
policy plan over the globe for building green planet again as soon as possible, avoiding
further environmental degradation. Let's reduce fossil fuel usage, stop cutting trees by
planting more, ban the Rampal project, together heal the world, make it a better place."
Comments emphasized the need to act locally, for instance, stopping the Rampal coal-based
power plant, located near the Sundarbans, a UNESCO World Heritage Site, which is causing
an adverse impact on the Sundarbans' biodiversity and ecological conditions. The current
administration built the power plant, ignoring feedback from both environmental experts and
the masses. There was a collective effort to stop the project, but it went forward anyway.





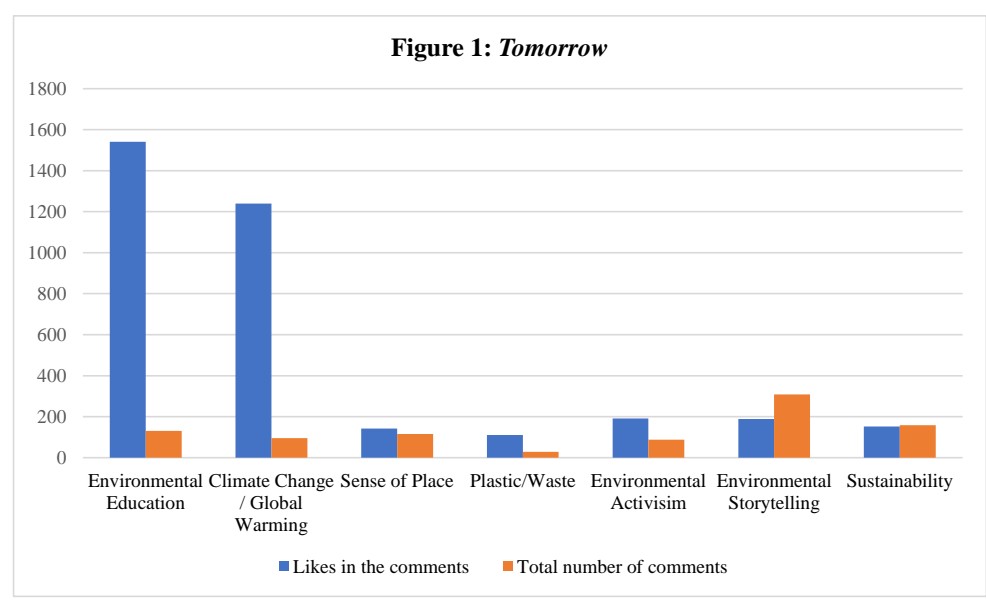



Figure 1 shows the relationship between the total number of comments by topic in the
sample alongside the total number of likes for that topic. A high like bar indicates many likes,
the easier of the two participatory actions. Total comment bars tend to be lower as
commenting is harder; the higher an orange bar, the greater the desire to perform deeper
participation. The ratio between the two shows how well comments are received.
Public comments demonstrate two major demands. First, this film should be
disseminated more widely, including being translated into English and other languages. (The
film was indeed later translated into other languages.) People from outside Bangladesh
should know that the Bangladeshi film industry can make a film that meets international
standards, and people from everywhere should be aware of climate change and become active
in civic discourse to hold their leaders to accountable. Second, common people should
engage in environmental activism and take peaceful civic action. The film suggests imposing
taxes on fossil fuels and investing in renewable energy, solutions echoed in the comments:





the public—individually and collectively, locally and globally—want to create and contribute
to a broad environmental movement.

The commenters exhibit a sense of urgency to create a sustainable planet Earth, as

well as their local environment. The tax solution to climate found in *Tomorrow* is also found
in comment analysis: online commenters want to create a movement to combat climate
change and plastic production. This shows that the calls to action given by the film in both its
visual and narrative storytelling were effective in at least inspiring viewers to comment on
their desire to act.

The public reactions to *The Lorax*, based on 1000 tweets, are similar to, yet distinct

from, those to *Tomorrow*. For example, "The Lorax is a cinematic masterpiece" and similar
comments show the widely shared opinion that the film had a high production value and was
enjoyable. As was the case with *Tomorrow*, the public reaction was also emotional for *The*
*Lorax*—the public cannot tolerate injustices and environmental destruction, even in fiction.
Table 2 and Figure 2 demonstrate the major environmental discourses coded from the public
comments, and the counts of these comments.[4]

---

[4] Netlytic derived Twitter comments excluded "likes," thus the exclusion compared to Table 1.



**Table 2: Examples of Twitter Comments from *The Lorax***
***(Arranged by total number of comments)***

| Discourse | Total number of comments | Examples of comments |
|---|---|---|
| Environmental education | 21 | "y'all are getting literal degrees and careers still not believing in climate change. my 2 year old sister understands climate change and all she did was watch the Lorax." |
| Sense of place | 31 | "I am the Lorax and I speak for the trees Save the Amazon, or I'll break your knees." |
| Climate change | 39 | "#Earth #water The biggest issue of our time #climatechange #unless 'Unless someone like you cares a whole awful lot, Nothing is going to get better. It's not.'" |
| Plastic/waste | 41 | "i am the lorax and i speak for the trees litter again and i'll break your fucking knees 😁" |
| Environmental activism | 92 | "@JohnBrennan @tedcruz He read Green Eggs and Ham on the Senate floor. I read the book to my 3 children countless times. Can I be a Senator from Texas now? The Lorax is the finest Dr. Seuss book and when Senator I will read that on the Senate floor." |
| Environmental storytelling | 111 | "It is a very deep and inspirational movie … I truly think it should have won movie of the year in 2012. It has changed my life for the better. This post is not satire." |
| Sustainability | 116 | "The Lorax are we planting more trees — In sha Allah, many more 🎇 ❤️ https://t.co/evJu2P0iIb." |


The most frequent subject of public comments relates to the multifaceted issue of
sustainability. The audience knows that online activism can be an effective tool for creating
political pressure and social action. An example of a commenter calling for social action is as
follows:
"As the wise Lorax once said "Unless someone like you cares a whole awful lot,
nothing is going to get better. It's not." On Saturdays, join us for a Self-Guided Beach
Cleanup. Make a difference in as little as two minutes. https://t.co/lUpXKlmoy3
#volunteer"
Perhaps the environmental storytelling used by the films is why commenters actively ask for
initiatives aimed at reducing the plastic impact.



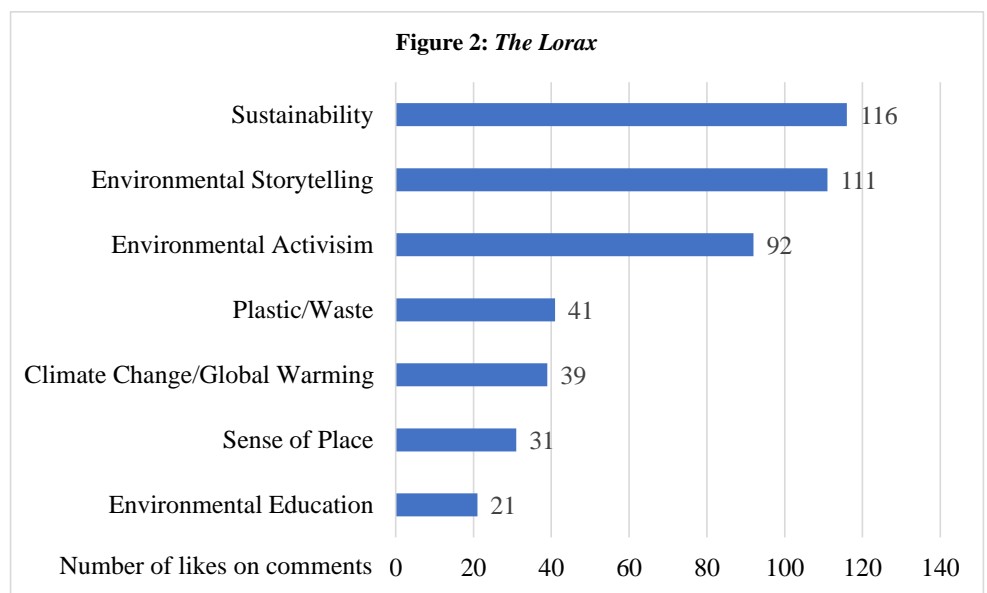

**Figure 2:** *The Lorax*


Figure 3 compares the terms found in the environmental discourse seen within the
public comments. Since *Tomorrow* has more analyzed comments (1510), it appears higher
than *The Lorax* (1000) in all discourses when directly comparing raw data. Figure 4 therefore
compares the percentage of comments by coded subject.
Today's academic environmental activism draws inspiration from Thoreau, Muir,
Leopold, and Carson, among others, with this academic discourse indirectly influencing ideas
found in public activism through the broader environmental movement. *Walden* (1854) laid
the foundation of modern-day activism because Thoreau coexists with nature. Muir's
establishment of the Sierra Club and encouraging ordinary people to explore Yosemite
Mountain shows activism. Leopold (1986) considers the land as a teacher and emphasizes the
restoration of land is an enduring example of environmental activism. Leopold (1986)
remarks, "A thing is right when it tends to preserve the integrity, stability, and beauty of the
biotic community. It is wrong when it tends otherwise." Carson's (1962) *Silent Spring* is
enduring because it shows women's activism contrary to men's, and it demonstrates her bold
statement against the patriarchy, which is responsible for pesticides and insecticides. In





addition, like *Silent Spring*, Wolfe (2008) argues that *The Lorax* warns of a present danger
and a rapidly approaching future. Comment activity demonstrates that the public also relates
race, ethnicity, and gender with environmental activism. Some comments contend that *The*
*Lorax* is racist and sexist because the Lorax speaks for only for certain trees, as exemplified
by the comment "Quick question: is the Lorax racist against certain trees? He just seems like
the type"[5] and some believe Audrey should have been the protagonist instead of Ted. Despite
the fact that online commenters presumably do not often have backgrounds in academia, it is
notable that a casual informal understanding of intersectionality is sometimes seen within the
comments. In the United States, campaigns about environmental justice have been
historically intertwined with race, class, and gender. For instance, environmental historian
Nancy Unger (2012) has written about how women often interact more closely with their
local environment than men do. Similar to the work of Unger, Afro-American cultural
geographer Carolyn Finney (2015) addresses environmental justice in *Black Faces White*
*Spaces*. Finney reviews the history of African American engagement with the mainstream
environmental movement from the early 1900s to the present. Finney focuses on how African
Americans are excluded from the environmental justice movement, but she espouses the
human experience of the story. Public comments tend to deconstruct the hegemonic racist
elements, if informally.

---

[5] Username expunged.





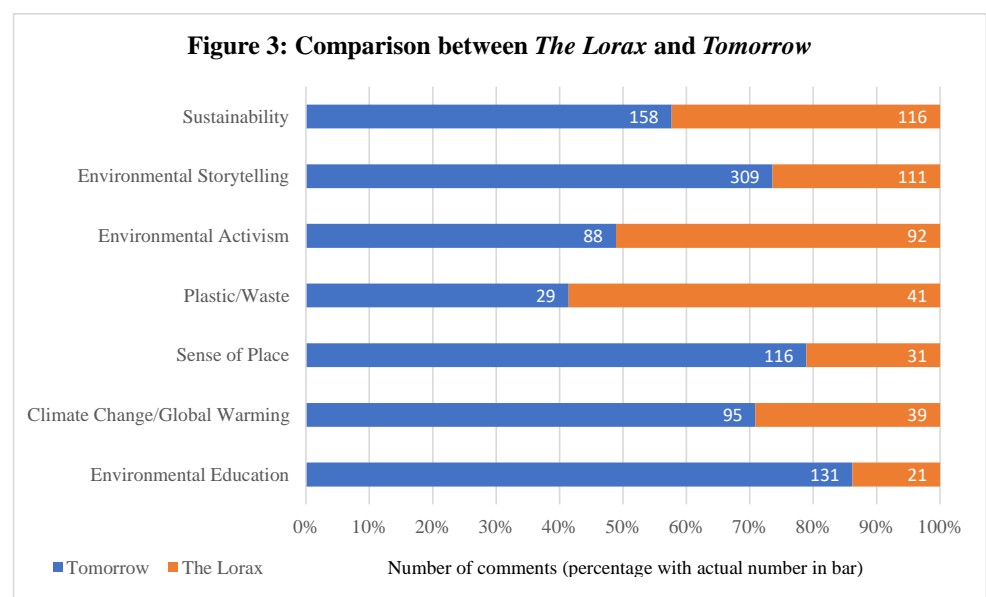



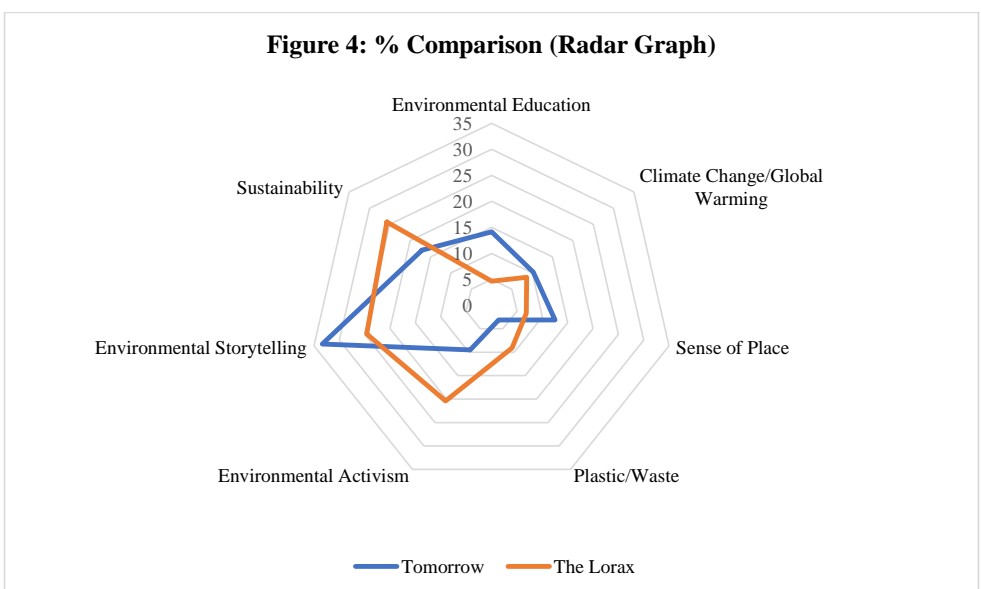



When analyzing the discourse of any text, through skip-grams, bigrams, or n-grams, a
word association network prioritizes word-by-word analysis. It is important to note that
methodologically speaking, a single analysis of just one of these graphs could be highly



misleading – they must be interpreted together, and with context of the films, to avoid
making inferences which are not based in reality. For example, methodologically, the value
of "Lorax" appearing in a word frequency table so much should be discounted, because the
use of the word could plausibly refer to the film, the book, the old television episode, or the
character himself. However, by viewing the other graphs, enough context can be gleaned to
provide cautious insight.

For *The Lorax*, the word-frequency table demonstrates the top word counts of the

selected tweets, in which the word "Lorax" appears in nearly 800 tweets while "Once-ler,"
the least common term on the list, is in many fewer tweets. However, the count for the word
"Lorax" is included below to provide greater context for a later skip gram analysis. Because
of the discounting of the word "Lorax," the most significant term here is perhaps "like"
which while not a perfect indicator, generally indicates positive sentiment in conjunction with
the relatively high-ranking word "good." This is especially noteworthy when compared to the
lower ranked word "bad" (which may also be affected by its heavy use in the fan favorite
song "How bad can I be?"). The word "trees" appears to be relatively highly ranked,
indicating strong environmental sentiment in viewers.[6] Finally, the pair of words "watch" and
"watching", while individually ranked lower on the graph, would jointly rank higher, and are
often used in comments to indicate personal interaction with the film itself. One example
comment illustrating this follows: "@[7] Hey lol, wanna watch the lorax together 👉👈"

---

[6] The use of the Truffula tree as a movie plot point could contribute to this word being highly ranked, however since the Truffula Tree is a fictional proxy for the overall environment, this inference is appropriate.
[7] Username expunged.




**Figure 5: Top word count for *The Lorax***

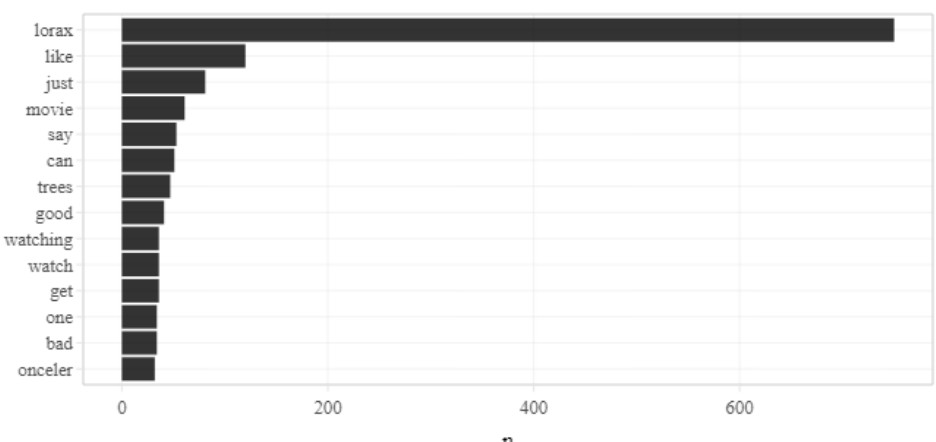


**Figure 6: Skip-gram for *The Lorax***

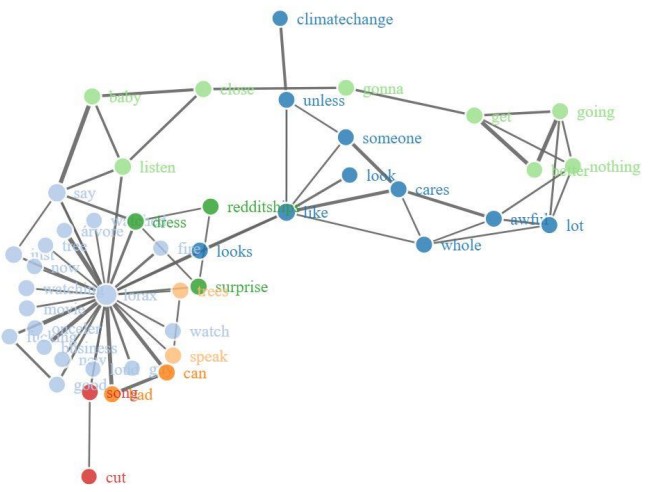


For *The Lorax*, the skip-gram count network exhibits the following word chunks. This
skip gram is centralized around "the Lorax" with a cluster influenced by the phrase "unless
someone cares a whole awful lot." While in the previous example, the term "Lorax" should
be discounted, here the related words show a roughly even split between discussion of the





movie itself or the character to these ideas, making it more useful within this context
compared to the previous figure. For the roughly half of instances which emphasize the
narrative of the film over the movie itself, this is linked with the environmental discourse
regarding climate change. Other notable words are revealed by the skip-gram word count,
including "redditships," referring to offsite discussion of non-canonical romantic relations
between characters, and "dress," which can perhaps be explained by the distinctive clothing
worn by the characters. In the skip-gram, the word "dress" is directly connected to the word
"redditships", indicating a close attachment to characters within the narrative, and is also
located (albeit indirectly) in close proximity to the word "looks" indicated an emphasis on
aesthetic value judgements. Perhaps factors such as fashion and the aspect of potential
romance between characters are also something to be considered when designing new
environmental media in order to improve audience engagement.

**Figure 7: Top Word Count for *Tomorrow***

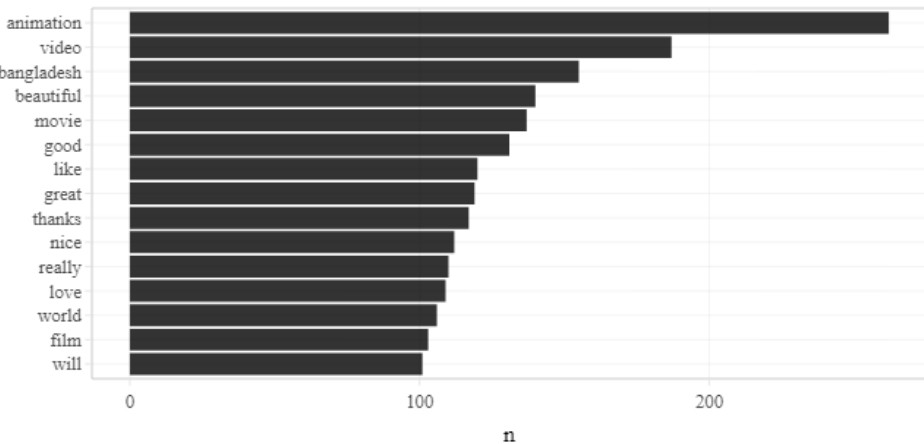




**Figure 8: Skip-gram for *Tomorrow***

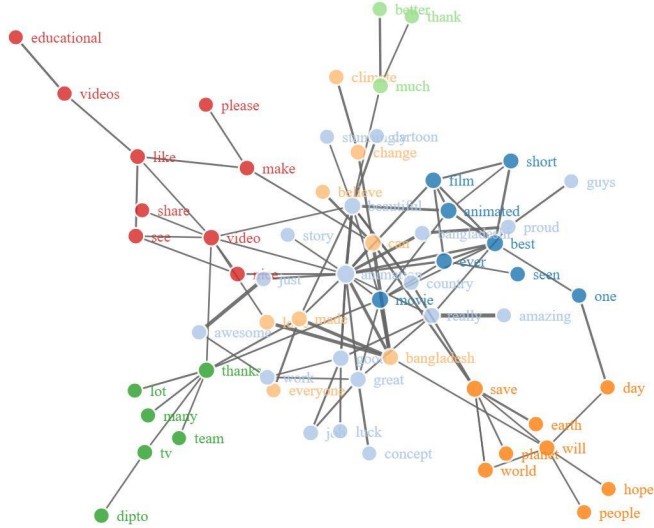

For *Tomorrow*, the word-frequency table shows that "animation" is the most frequent

word that appeared, though this is perhaps affected by the discussion of the film industry in
Bangladesh.[8] Many of the highly linked words in the skip-gram demonstrate the positive
sentiment of the film and its content generally, such as "awesome", good, "great", etc. It is
notable that even though the word "Bangladesh" is ranked so high in the top word count, the
word "world," while relatively low ranked among the top words, still demonstrates a global
consciousness among the online commenters. This is especially illustrated by the skip-gram,
where the word "world" is closely associated with the words "hope," "save," and "planet."
Further analysis of the skip-gram count network for *Tomorrow* reveals a more decentralized
network when compared to *The Lorax*. This is potentially influenced by the lack of a specific,
unifying catchphrase in *Tomorrow*, whereas the central "unless" catchphrase of *The Lorax*
comprises a significant wordchunk.

---

[8] A number of online comments take nationalistic pride in a progressing film industry in Bangladesh, using *Tomorrow* to exemplify advancing standards in domestically produced computer animated films.



### 5. Environmental discourse analysis


How film characters deal with environmentally catastrophic issues is part of what
viewers imitate and can be influenced by in a film. These two films exhibit several major
environmental themes, including the concept of unspoiled nature, the sense of place,
pollution, deforestation, and land erosion.
Environmental discourse analysis has been adapted and developed from several
branches of social sciences, primarily anthropology, and is therefore inherently multifaceted.
Adrian Peace (2018) explains that "academic disciplines go about their interrogation of
discourse in different ways" (p. 415), but in general describes discourse as "specific ways of
talking about particular environments and their futures" (Mühlhäusler and Peace, 2006, p.
458). A social anthropologist, for example, "become[s] familiar with the natural discourses
local people draw upon to describe environments of greatest significance to them" and in this
way contributes to environmental discourse analysis by highlighting environmental discourse
on a local level (Peace, 2018, p. 415). Peace (2018) is a social anthropologist, but there are
many historians, political scientists, or communication studies researchers who examine
power abuse, inequality, and other significant concerns within the social and political
environment. The many discourse analysis techniques cannot be summarized in this brief
space, but all approaches, at least to an extent, view language as a social practice and
discourse as pertinent to the broader social order. This paper avoids the debate over the
(dis)similarities of environmental ideologies and environmental discourses to focus on
empirical findings. Mühlhäusler and Peace (2006) underscore that "[m]uch environmental
discourse elaborates the theme that human actions are detrimental to the survival of
humanity" (p 461). My analysis correspondingly highlights the irreparable damage that
humanity is contributing to the environment which viewers witness within the selected films.



Mühlhäusler and Peace (2006) speculate that it is yet unknown how much the
environmental discourses and metadiscourses improve the condition of the environment (p.
457). Environmental discourse analysis can show which narratives instill feelings of
hopelessness, apathy, and inaction in viewers. Conversely, environmental discourse analysis
may highlight narratives about environmental issues and matters of environmental justice that
give the viewer a manageable sense of alarm, spurring them to act before it is too late. In
"Envisioning A Sustainable World," Sustainability scholar Donella Meadows (1994) regrets,
"Whatever the reason, hardly anyone envisions a sustainable world as one that would be
wonderful to live in" (p 2). She is hopeful nevertheless, "I have noticed, going around the
world, that in different disciplines, languages, nations, and cultures, our information may
differ, our models disagree, our preferred modes of implementation are widely diverse, but
our visions, when we are willing to admit them, are astonishingly alike" (1994, p. 4). Two
different movies from two different parts of the world with two different senses of place both
demonstrate a singular desire to save the world from environmental catastrophes.

**6. Environmental catastrophe**
*The Lorax* takes a social constructionist view of nature as the film explains that the
trees, the forests, have agency, but must act within a framework established by mankind—an
anthropocentric view. The Lorax, the guardian of the forest, thus establishes a space to
advocate for the rights of nature. The Lorax's proclamation in the opening scene, "I am the
Lorax, I speak for the trees," (0:00:56) establishes the role of the Lorax in representing nature
more broadly. The tone of the film is set by the deceptively bright city of Thneedville, set
against a foreboding sickly purple dawn. This city is one of artificiality, in which every entity
is made of artificial products: "a town without Nature, not one living tree" (0:01:24). In this
city, trees are made of plastic and their colors can be changed by clicking remote buttons.



Environmental pollution in the film is often implied through use of plastic, and the exclusive
use of synthetic materials instead of those found in nature. The artificiality of Thneedville
constitutes a major crisis in the film. Thneedville society takes capitalist pride in
commodifying nature: O'Hare informs Ted, "I make a living selling fresh air to people"
(00:31:19). Ted's search for an original tree is a business threat to O'Hare's company. Moore
(2016) explains the intimate relationship that exists between children, consumer culture, and
commercial media in the United States. Moore (2016) shows that both "the news and
entertainment industries reveal that the way Hollywood treats a subject like the environment
is not an exception to the rule; instead, the consistent subjugation of environmental concerns
is part of a broader capitalist logic in a concentrated market" (p. 5). This also connects to
real-life industrialists, as in when Frankfurt School critics Max Horkheimer and Theodor
Adorno (2007) discuss capitalist social structures, arguing that material identities are assigned
to nonmaterial cultures (perhaps also natural resources), commodifying them into the
products from which capitalists could profit. These natural resources are manufactured,
bought, and sold like a commodity. Environmental historian William Cronon (1996) has also
described the impact of nature as commodity not just in American culture and landscape but
in the entire planet Earth for centuries.

The excessive use of plastic and artificiality are symbols of late-stage capitalism. In

that regard, the opening song's lyrics stress the phrase "brand new" that references that we
live now in an advanced capitalist society which fetishizes consumerism: "If you put
something in a plastic bottle, people will buy it" (11:24). Commodification increases when
natural entities are treated without respect with some exceptions. The film implies that
Truffula trees are valuable and a positive, desirable asset, because by providing food, shelter,
and oxygen, Truffula trees help reduce environmental threats.



*Tomorrow* also presents the idea of nature, but it is not a socially constructed nature,
nor a nature that is soothing and tranquil. Rather, it emphasizes that reckless behavior from
humanity not only damages the environment, but also makes nature uninhabitable for
humans. Irresponsible human actions not only damage the environment, but they also make it
more vulnerable to future damage. The village in *Tomorrow*, unlike Thneedville, is not
artificial, yet its people lack a sense of environmental consciousness just as in *The Lorax*,
until Ratul's father joins the conversation about the land erosion with the people. Their
conversation and the conversation between Ratul and the Old Man of the Winds change their
attitude—they gain an understanding of nature which make them proactive in slowing down
the unfolding disaster and envisioning—literally showing the audience—a future full of hope.
Such a positive narrative work against the idea that it is too late to act to prevent catastrophes.

Plastic waste is another environmental catastrophe on its own, which additionally is a

contributing factor to global warming, as plastic production and transportation require fossil
fuels. *The Lorax* shows the audience the impact of waste and wanton consumption on the
environment. The Lorax demonstrates that the process of wanton cutting down trees and
making clothes (fantastical knitting) out of it as a wasteful practice. But the Once-ler,
considers the result of this tree-cutting and knitting process "revolutionary." The product has
a multitude of uses, and the audience may be inclined to agree at first, enhancing the impact
of this cautionary tale. "Whoa," is Ted's reaction when he steps out of the walled Thneedville
and sees the industrial waste. Thneedville produces a lot of waste but has no policy regarding
waste management beyond hiding it from public view; this is a reminder that the whole world
suffers from waste management policies that are effectively wishful thinking and likely
encourages the viewer to consider the impact of the industrial society fuelled by their own
wanton consumption.

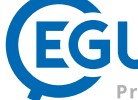

Industrial waste is also a critical theme in *Tomorrow*. In the dream, when the Old Man
of the Winds takes Ratul on a tour of the world, Ratul notices chimneys spewing greenhouse
gases in the atmosphere. The Old Man of the Winds instructs Ratul that coal needs to be
replaced as a source of energy.
Both *The Lorax* and *Tomorrow* encourage the viewer to foster a desire to protect
nature, albeit in different ways. *The Lorax* fosters a protective desire through Audrey's and
Ted's quest for Truffula trees, Grammy's Indigenous sense of conservation, and the Lorax's
mission to speak for trees. Although Wolfe (2008) focuses on Dr. Seuss's book, Wolfe's
observation that, "[. . .] nature is elevated from inferiority to a form of divinity" is germane in
the context of film (p. 14). *Tomorrow* fosters love for nature by creating an awareness about
climate change and biodiversity.

**7. Environmental storytelling: sense of place**
These films deal with both place and displacement, important concepts in
environmental education. "The integration of place into education is important," writes
sustainability scholar David Orr, as "knowledge of place where you are and where come from
is intertwined with knowledge of who you are. Landscape, in other words, shape mindscape"
(2013, p. 93). These films use storytelling show how human beings and animals are
displaced.
Storytelling is an important element in combatting large-scale problems such as
climate change. Stories lead to greater emotional attachment than raw data does. In looking at
the impact of stories, I return to Jonathan Gottschall's (2012) statement, "we are, as a species,
addicted to story. Even when the body goes to sleep, the mind stays up all night telling itself
stories" (12). Cherokee author Thomas King's (2003) statement, "The truth about stories is
that that's all we are" (2) or environmental historian William Cronon's (1992) statement





inspired by Graham Swift that human beings are "storytelling creatures" underscore the
importance of storytelling. But humans are not the only storytelling creatures; other animals,
plants, spirit beings are the storytelling creatures too. The tree, the land, the other non-human
entities have agency, and they are storytelling creatures too, as demonstrated in the films. In
*The Lorax,* although of course somebody else has to speak for those trees, they nonetheless
have agency, and in *Tomorrow*, the Old Man of the Wind is not human but rather a spiritual
entity. By incorporating non-humans into storytelling, these movies help combat human
supremacist attitudes, by showing that man cannot stand alone against environmental
collapse.

Movies cover important environmental features in the form of storytelling discourse,

which also encompasses the field of storytelling discourse aimed at children. Dolores Subia
BigFoot and Megan Dunlap (2006) note that "[s]tories give reason to the overall scheme of
things" (p. 134). *The Lorax* and *Tomorrow* carry an environmental storytelling tradition to
teach children a sense of place through stories (animated films are often aimed at children,
and teach both children and the parents; if children miss out anything, the parents can pick it
up). BigFoot and Dunlap (2006) suggest that "Parents, grandparents, and other relatives used
stories to help children understand their place in the world and how they could show their
gratitude for their existence" (p. 135). This is evidenced in the social media analysis, where
one commenter stated "my 2 year old sister understands climate change and all she did was
watch the Lorax"

Both films have a simple environmental storytelling trajectory, but that simplicity is

grounded within the place of each respective culture. *Tomorrow* focuses on a specific place
along the coastline of Bangladesh; *The Lorax* is a fantasy place that could be anywhere and
nowhere. If places are ecological and cultural, I would argue that the sense of place is linked
to the art of storytelling, ultimately linked to education and pedagogy. Orr (2013), for



instance, demonstrates the nexus between place and pedagogy. Orr's understanding of place
as an educational tool emerges from Thoreau's *Walden* (1854) (to be exact, "*Walden* is a
model of the possible unity between personhood, pedagogy, and place") and conservationist
Leopold's (1986) philosophy of "man as a biotic citizen." Although non-human entities are
appropriated for our use, *Walden* (1854) emphasizes natural entities in a way that could help
contemporary culture be more sustainable, such as in issues like bottled water compared to
tap water. A similar perspective can be seen from Leopold, who draws us across time and
space by introducing ideas like the "land ethic" and asking human beings to think "like a
mountain." These philosophies should be highlighted with a greater emphasis in popular
culture. Beyond Thoreau's and Leopold's philosophies, I would add the controversial
philosopher Martin Heidegger's idea of nature as "physis," as discussed by Timothy Clark
(2011), in conjunction with Orr's idea about place, which is useful and has awe, splendor,
beauty, majesty, and a force. Heidegger's "physis" is visible when Orr suggests
that *Walden* is a dialogue between a human being and a place. This dialogue refers to a sense
of love with a cultural identity for the place. By and large, these philosophies demonstrate
how place plays a role in our moral and psychic transformation. In *Tomorrow*, although
commoners lack an academic or formal understanding of place consciousness, they
eventually show the unity and a sense of belongingness needed to protect and preserve their
local place. In *The Lorax*, the not-real place still demonstrates influences from its Californian
creators of 2012, such as general heightened concern over environmental catastrophes like
the 2011 Fukushima disaster in areas on the Pacific. In the film, a child begins glowing a
radioactive green as he sings "I just went swimming, and now I glow!" (0:03:03). Stories
with connections to place are important as "[s]tories can give children a sense of belonging to
their family, community, and tribe, and this can instil a sense of purpose, identity, and hope.
Stories could be an extremely positive force in the life of children" (BigFoot and Dunlap,




2006, p. 5). *The Lorax* and *Tomorrow* share the spectrum of life stories: as evidenced by
online comments such as "local places are getting destroyed because of global places," these
stories can create a compelling connection between the storyteller and listener/spectators.
These two films are similar in that common people within the films are engaged to love their
local places. Initially, the Once-ler's family is not respectful of the local place and
environment, but when the Once-ler gives Ted a seed to make the local place abundant with
trees, the local place matters. People rally around Ted for planting the seed, although they had
almost been convinced otherwise by O'Hare's deceptive speech. In the same way, the
common people start a movement to save their village from climate crisis in *Tomorrow*.

Storytellers can create a sense of connectedness with the stories. The Once-ler and

Ted's grandmother serve as the role of storyteller. Granny initiates the storytelling session,
but she sends Ted to the Once-ler for firsthand experience. The Once-ler starts sharing the
story with "it all started a long time ago" (00:16:45). The Once-ler's starting cue gives us a
sense of hearing a "once upon a time" story. He later uses the phrase "a long time ago," at
least three times, hinting that the environmental destruction on Earth started a long time
before. As a storyteller, the Old Man of the Winds, in *Tomorrow*, appears in Ratul's dream
and blames him for the deplorable condition of the planet. He takes Ratul away with him to
show the cruelty of people on nature: factories are emitting fumes and the use of fossil fuels
are resulting in air pollution, the greenhouse effect, and related human eco-sicknesses.

**8. Environmental education: sustainability**

Today, many animated films are incorporated into educational curricula because of

the impact they can have. *The Lorax* and *Tomorrow* are ideal candidates to be educational
tools for children as these films visually show (rather than just tell) fundamental



environmental problems and potential solutions. Both films can also be a platform to teach
children about preserving nature.

*The Lorax* and *Tomorrow* promote a world where sustainability and environmental

consciousness are prioritized over reckless economic and technological development. Both
films critique capitalism for setting society down a path of self-destruction. When Ted leaves
the town in search of the Once-ler, Ted is being watched on his way out by the corporate
enforcers of O'Hare, who report on anything that threatens their industrial progress. Progress
is the main goal for Thneedville's people. The Once-ler's mother rebukes him for not being
"successful." The Once-ler thus starts changing the world, but the spell of capitalism does not
fool the Once-ler forever, as he eventually recognizes the monstrous effects of unchecked
capitalism. In *Tomorrow*, Ratul learns about the impact of capitalism when he travels with the
Old Man of the Winds who shows him the advanced capitalist societies which are least
sustainable (even though some pretend to be). Ratul becomes conscious of the negative
aspects of the socio-political-economic nexus of capitalism, but he cannot remain free from
it. The films show the audience through visual storytelling what the consequences of life in a
capitalist society that reveres progress and success, and, later, how the lives of the people
within that society improve when environmental consciousness triumphs over the
commodities market.

The films both suggest that the destruction caused by unrestrained capitalism may be

averted through action. In *The Lorax*, Ted brings meaningful change by helping begin to
restore the environment. *Tomorrow* also offers solutions, such as imposing taxes on fossil
fuels and implementing green energy around the world. *Tomorrow* asks its audience "Are you
with us?" (20:55) and tries to create a sense of urgency to get its audience to act.

These films can help instill the idea that sustainability is more about actions rather

than just caring about nature. *Tomorrow* suggests an alternative to the present world by





showing a world occupied with environmentally responsible inhabitants; *The Lorax* suggests
a return to a more natural environment as an alternative to an artificially lavish life. These
alternatives are designed to preserve nature. It is important to note that these films do not
reject societal progress outright, but they oppose development rooted in industrial toxic
consumerism which can cause the displacement of millions of people and the extinction of
species.
Perhaps the most important characteristic for any educational tool is to leave a
discursive space. These films question their surroundings, the human interference with our
environment, the inevitable consequences of such interference, and they provide examples of
a remedy. *The Lorax* suggests it is "not too late"; that is, if people give up their
anthropocentric attitudes and seek harmony between nature and human, their doom can be
prevented. *Tomorrow* shows the need to be prepared for a calamity that cannot be evaded, but
also shows a glimmer of hope. It endows the audience with agency, when it tells Ratul, the
audience surrogate, "This is your planet, you have to find out the answer" (21:54). These
films leave unfinished tasks to be comprehended and finished by the audience.
These films also offer pedagogical opportunities because they convey their messages
through non-traditional formats such as social media, humor, song/rhyme, satire, etc. For
example, the song "How Bad Can I Be?" in *The Lorax* provides insight into the greed-driven
soul who avoids caring for a few trees in the desire to make money. In *Tomorrow*, there is the
presence of social media. Ratul starts campaigning on Facebook about fossil fuels from a
local place, and he receives global responses, as people from around the entire world protest.

**9. Conclusion**
Using environmental discourse analysis to understand how discourses about climate
change and sustainability, to list a few, are changing, is an important task. This is



acknowledged in the literature—Elliott and Cullis (2017) have written, "the humanities
should be more confident and vocal in addressing climate change" (p. 15). Although the
number of creative works on climate change is increasing, their growth is not as substantial as
the increase in risks we are encountering. Heise (2008) argues that "climate change poses a
challenge for narrative and lyrical forms that have conventionally focused above all on
individuals, families, or nations, since [climate change] requires the articulation of
connections between events at vastly different scales" (p. 205). Although it is challenging, the
most powerful environmentally driven artwork and films focus on local, regional, and global
riskscapes. The combination of these different scales described by Heise can be tricky, but
these films show it can be done. In *Tomorrow*, local action leads to global change, and in *The*
*Lorax,* action within Thneedville leads to improvements in the lands outside the city.

In *The Lorax*, as time passes, new trees begin sprouting, animals return, and the

repentant Once-ler joins the Lorax, everything in its proper place. The film ends with the note
that unless someone comprehends the awful consequences that awaits us and takes prompt
action, "nothing is going to get better." *The Lorax* seeks to promote ecological awareness
among people showing the repercussions of their deeds "unless" they start taking care of the
environment. Film critics say that *The Lorax* is too political or scares children from the
environment by giving them "ecophobia" (Potts, 2019). Yet some are more hopeful, such as
critic Deidre Pike (2012), who deems *The Lorax* a "'dialogic enviro-toon' not presenting a
subject merely for entertainment but creating a safe zone for exploration of environmental
facts, ideas, images, and perspectives" (p. 13). Public commenters generally seem to agree
with Pike, and do not seem hindered by the message of *The Lorax*. Rather than them being
too political and ecophobia-inducing, I would argue *The Lorax* and *Tomorrow* have the
power to inspire the next Greta Thunberg in households around the world.



The uncertainty with which *Tomorrow* starts is a recurring theme throughout its entire
runtime. The film ends with a note of hope which environmentalist Bill McKibben (2019)
praises in his tweet saying, "it never blinks at the horrors in store, but refuses to give up
hope." Human beings are driving the great sixth mass extinction, but there is still time to take
initiative—a sentiment demonstrated in both the environmental discourses in the content of
these films and the public reactions. We need creativity, imagination and hope to face the
environmental crisis. The environmental discourse analysis of these films and the public
comments symmetrically convey the message: nature is on the brink of disaster in both films,
nevertheless both give the audience hope for the future.
**10. Ethical Statement**
Hereby, I, Mohammad Mizan-Rahman, consciously assure that for the manuscript "Unless
someone like you cares a whole awful lot, nothing is going to get better": An environmental
discourse analysis of animated films *The Lorax* (2012) and *Tomorrow* (2019), the following
is fulfilled:
1) This article is my own original work, which has not been previously published
elsewhere.
2) The paper is not currently being considered for publication elsewhere.
3) The paper reflects my own research and analysis in a truthful and complete
manner.
4) All sources used are properly disclosed (correct citation).
5) The paper did not receive any funding from anywhere.
6) I only include data that has been processed in ways that do not identify individual
users, typically by describing the content of tweets/YouTube comments and/or citing
fragments that have generic wording in order to comply with recent ethical guidance
on the handling of social media data (Townsend and Wallace, 2016).
Date: January 10, 2023
Author's name: Mohammad Mizan-Rahman





**Works cited and consulted**
BigFoot, D. S., & Dunlap, M. (2006). Storytelling as a Healing Tool for American Indians. In
T. M. Witko (Ed.), *Mental health care for urban Indians: Clinical insights from*
*Native practitioners* (pp. 133–153). American Psychological Association.
https://doi.org/10.1037/11422-007
Caraway, K., & Caraway, B. R. (2020). Representing Ecological Crises in Children's Media:
An Analysis of The Lorax and Wall-E. *Environmental Communications*, 14(5), 686-
697. https://doi.org/10.1080/17524032.2019.1710226
Clark, T. (2011). The Inherent Violence of Western Thought? *The Cambridge Introduction*
*to Literature and the Environment*. Cambridge: Cambridge University Press.
Cronon, W. (1996). Introduction: In Search of Nature. *Uncommon Ground: Rethinking*
*Human Place in Nature*. New York: W.W. Norton & Co.
Elliott, A., Cullis, J. (2017). The Importance of the Humanities to the Climate Change
Debate. In: Elliott, A., Cullis, J., Damodaran, V. (Eds.) *Climate Change and the*
*Humanities*. Palgrave Macmillan, London. https://doi.org/10.1057/978-1-137-55124-
5_2
Finney, C. (2015). *Black Faces, White Spaces: Reimagining the Relationship of African*
*Americans to the Great Outdoors.* The University of North Carolina Press: Chapel
Hill.
Garrard, G. (2014). *The Oxford Handbook of Ecocriticism*. Oxford University Press.
Gottschall, J. (2012). *The Storytelling Animal: How Stories Make Us Human.* Boston/New
York: Houghton Mifflin Harcourt.
Gruzd, A. (2020). Netlytic: Software for Automated Text and Social Network Analysis.
Available at https://Netlytic.org/index.php. *Nostalgia Critic Commentary: The*



*Lorax.* https://channelawesome.com/nostalgia-critic-commentary-the-lorax/

Heise, U.K. (2008). *Sense of Place, Sense of Planet: The Environmental Imagination of the*

*Global.* New York: Oxford University Press.

Horkheimer, M. & Adorno, T. (2007). "The Culture Industry: Enlightenment as Mass

Deception." In S. During (Ed.), *Dialectic of Enlightenment. The Cultural Studies*

*Reader*. London and New York: Routledge.

Howarth, C & Anderson, A. (2019). Increasing Local Salience of Climate Change: The Un-

tapped Impact of the Media-science Interface. *Environmental Communication*, (13)6,

713-722. DOI: 10.1080/17524032.2019.1611615

Hulme, M. (2014). Why We Disagree About Climate Change: Understanding Controversy,

Inaction and Opportunity. In Greg Garrard (Ed.), *The Oxford Handbook of*

*Ecocriticism*. Oxford University Press.

King, T. (2003). *The Truth About Stories: A Native Narrative*. Toronto: House of Anansi

Press.

Kluwick, U. (2014). Talking about Climate Change: The Ecological Crisis and Narrative

Form. In Greg Garrard (Ed.), *The Oxford Handbook of Ecocriticism*. Oxford

University Press.

Leopold, A. (1986). *A Sand County Almanac and Sketches Here and There*. Illustrated by:

Charles W. Schwartz. London/Oxford/New York: Oxford University Press.

Meadows, D. (1994). "Envisioning A Sustainable World."

https://donellameadows.org/archives/envisioning-a-sustainable-world/

Moore, E. (2016). Green Screen or Smokescreen? Hollywood's Messages about Nature and

the Environment. *Environmental Communication*, 10(5), 686-

697, DOI: 10.1080/17524032.2019.1710226



Mühlhäusler, P. & Peace, A. (2006). "Environmental Discourses." *Annual Review of*

*Anthropology*, (35), 457-479.

McKibben, B. [@billmckibben]. (2019, December 29). *Loosely inspired by Dickens, this*

*Bangladeshi animated tale about climate change is wonderfully done. It never blinks*

*at the horrors in store, but refuses to give up hope*. [Tweet].

Twitter. https://twitter.com/billmckibben

Orr, David W. (2013). Place and Pedagogy. *Ecological Literacy: Education and Transition to*

*a Postmodern World.* New York: State University of New York Press. 125-131

Peace, A. (2018). Environmental Discourse Analysis. *Environmental Humanities*. In Noel

Castree, Mike Hulme, and James Proctor (Eds.). Routledge, London and New York.

Pike, D. M. (2012). *Enviro-Toons: Green Themes in Animated Cinema and Television.*

Jefferson, NC: McFarland.

Stephanie, L & Stephanie, F. (2012). The Sustainable Humanities. *PMLA*, (127), 572-578
Renand, Chris. (2012). *The Lorax*. Illumination Entertainment.
Thoreau, H. D. (1854). *Walden*. Obtained from Project Gutenberg,

http://www.gutenberg.org/ebooks/205.

Townsend, L. and Wallace, C. (2016). Social Media Research: A Guide to Ethics. Aberdeen:

University of Aberdeen. Available at: www.gla.ac.uk/media/media_487729_en.pdf

(accessed 10 January 2023).

Uddin, M. S. (2019). *Tomorrow.* Kazi Media Limited.
Unger, N. (2012). *Beyond Nature's Housekeepers: Women in American Environmental*

*History*. Oxford University Press (Illustrated Edition).

Van Rossum, G., & Drake, F. L. (2009). *Python 3 Reference Manual*. Scotts Valley, CA:

CreateSpace. https://towardsdatascience.com/how-to-scrape-youtube-comments-with-

python-61ff197115d4



Wolfe, D. (2008). The Ecological Jeremiad, the American Myth, and the Vivid Force of

Color in Dr. Seuss's *The Lorax. Environmental Communication*. 2(1), 3-24.