# Peer review of ""Unless someone like you cares a whole awful lot, nothing is going to get better""

_EGUsphere, 2022_

## Referee Comment (RC1)

General comments

Thank you for submitting your work "Unless someone like you care a whole awful lot, nothing is going to get better": An environmental discourse analysis of animate films *The Lorax* (2012) and *Tomorrow* (2019)."  I found the comparison between the two films to be interesting, especially the analysis of the public comments and associated categories used to code them. A major strength of the paper is that it uses two movies from different cultures and contexts. As climate change is a global issue, looking at these themes from various perspectives is incredibly important, and I thought this paper did a great job of that.

While I think that this is an important paper with great potential, I think that there are some organizational changes that would make a big difference for readers navigating the content. There is a lot of analysis going on in this paper, and it can be hard to track what goes where and how the different pieces fit together. In fact, I think it could probably be separated into two papers: one for the discourse analysis and another for audience-commentary analysis. There are really great points made in both, but the way the paper is organized has them fighting against each other a bit.

In addition to the organization, I think that this paper could really benefit from streamlining the analysis throughout. There are so many scholars and ideas that are brought along to weigh-in on these two films, that a clear and consistent lens never materializes. For example, a paper that uses just the environmental scholars (Muir, Thoreau, etc) would be great. A different paper that uses a more Marxist lens (Horkheimer and Adorno) could also be great. Bringing together all of the theorists, though, dilutes the power of each since there isn't enough time to fully explore any one angle and make a focused argument. The great news is that I think this means that you have several great papers all wrapped up in this one paper, and that teasing them apart from each other could generate several much more focused arguments across different publications.

There is much to celebrate in this paper, and with some refocusing, I think it offers a great perspective on why movies like *The Lorax* and *Tomorrow* can make such a large impact on future discussions about climate and sustainability.

Specific comments

Here are some more specific recommendations listed by section:

Abstract
- While I thought it was great to identify in the abstract why these films were chosen, I would recommend adding a sentence or two early on to frame the problem or gap that this article fills. Giving a clearly articulated "why" we need this paper and what it might mean for environmental communication more broadly would give your audience early investment in the films that you are studying and their importance.

- The last two sentences could be strengthened by adding specificity. For example, how do these films shape the discourses of their audiences? What kind of large-scale environmental issues might films like these address?

Introduction
- While I appreciate the opening idea from Ursula Heise, I would recommend either adding more to the first paragraph or omitting it. I think you say the same thing in the second paragraph starting "Using environmental discourse analysis…" with a bit more power and specificity.
- A bit more information up front about why these two films were selected (this was in the abstract, but I think unpacking it a bit more in the intro would help your audience understanding the significance. Something I really appreciated about this paragraph was the focus and importance of a global perspective.

Methodology
- This section has a lot of great background information on what you did with the data and why. Where I think it could use some help is in organization. It feels like the first paragraph should perhaps switch with the second—I found myself wondering where the public comments came from in paragraph one, but that isn't explained until paragraph two. Moving from general to specific is one way to ensure that the audience can follow along without any information missing.
- I understand from your methodology section that you are using discourse analysis, but there can be a bit more attention to explaining what each step means (for example, how does your narrative analysis incorporate spatiality? What does that look like for audiences who may not be familiar with these methods?).

Film synopses
- I would recommend adding a few more sentences of context to the initial paragraph to concretely distinguish the movie from the book.
- Most of the paragraphs in this section strike me more as analysis than film synopsis. If I did not already know the story, it would be difficult to connect the text here to the storyline. I found myself a little bit lost trying to follow the ideas. The following paragraph has more plot details, but Audrey's character is painted against the general population of Thneedville, though there wasn't much background given on what that means (perhaps there is a paragraph of background missing)? There is analysis about who the Once-ler and the Lorax represent, but I think it would be difficult to understand without having seen the movie.
- I thought the synopsis of *Tomorrow* was much easier to follow, there was occasional analysis, but mostly storyline. I would embed a few more citations to credit the movie and the source attached to the budget, but this felt much more like a synopsis.

Analysis of public reactions
- I thought the overview of why public comments matter was great. When it comes to Table 1, my question is: are these the most liked comments in the whole sample? Or is

each comment the most liked example of the environmental discourse category it was coded with? I ask because it seems unlikely that the top answers were naturally occurring examples of the seven categories.

- I thought that the portions of analysis that paired specific comments with claims about their impact was well done.
- In Figure 1, the two bars draw the eye to compare the blue vs. orange, but I assume the goal is really to compare the orange bars with other orange bars and the blue bars with other blue bars. I would consider separating these into two visualizations.
- I found myself wanting more analysis of each of the seven categories. Organizing the analysis along these lines might make it easier to follow along and connect the text with the tables a bit more directly.
- The paragraph about Thoreau, Muir, Leopold, and Carson feel a bit more like background than analysis, and might work better in the introduction or framing of the paper. Similarly I think that the information about intersectionality was really great, but probably could have it's own section. I'm not sure I understand how it relates to the comparison figures that directly follow this information.
- The skip-gram in Figure 6 is interesting, but many of the terms in light blue overlap, making it difficult to read. I thought the analysis of the skip-gram in figure 7 with the connection between "world" and "hope" and "save" was well made.

Environmental discourse analysis
- This section felt like it should perhaps come before the last section, since it is similar in content to the synopses. Going back to discourse analysis felt a little bit jarring after the public commentary analysis. Also, it didn't really discuss the movies. It might be useful to fold it into the background or intro instead of have it as a separate section after the analysis of examples.

Environmental catastrophe
- Some interesting findings in here, but they feel like they link to ideas from way earlier in the paper.
- Some of these theorists, like Horkheimer and Adorno, are dropped in to the paper and moved on from very quickly, but their ideas deserve a bit more time and attention. I would consider focusing on one lens, rather than brining so much different scholarship to analyze the content. It is probably too much for one paper.
- I think these are all really good examples of how catastrophe is represented in the films but I'm not sure what the larger argument is.

Environmental storytelling
- Again, there are so many theorists whose ideas are listed, but without much substance to them. I would focus on one or two and use them to make a compelling argument about storytelling, which I think is an important theme that the paper picks up on.
- The paragraph that starts "Both films have a simple environmental storytelling trajectory" is another example of a paragraph that skims the surface of many different scholars without spending much time on any of them.

Environmental education
- Great appeal to include these films in education—I think that is a really solid argument that you can make in this paper.

Conclusion
- I think that you did a nice job of summarizing why these films are important and how they might help us reimagine a more sustainable future.
- I would recommend using the sources earlier in the paper. By the conclusion, most of the writing should be about next steps rather than adding new content.

Technical corrections
- Artwork can be singular or plural so I don't think it is technically incorrect, but I stumbled a bit on "they" standing in for artwork in lines 30-32. It could be a personal preference. Also, is artwork in this case meant as the movies you are working with? Those also feel slightly different to me, or like they could use slightly more explanation.
- I would consider incorporating sources to justify how you chose the number of comments somewhere in lines 88-97.
- Be sure to include sources for the background information regarding the movies (lines 113-114, for example).
- "ethic of care" perhaps? (239)
- I think "to explain" might build better construction with the first part of the sentence (242)
- "indicating" or "which indicates" (387)

---

## Author Response (AR1)

Hello,

Thank you for your valuable feedback! I have used it to polish and revise the paper into a better work. Your point of improving organization and streamlining is an important one, which I have incorporated into my revised version.

Introduction:

Per your suggestion, I experimented with expanding Ursula Heise, but ultimately it was too disruptive to flow, and so it was omitted in the first paragraph per your other suggestion.

I added more justification as to "why" these two films were specifically chosen.

Methodology:

Your direction to move from more general to specific here and to swap those two paragraphs greatly increased flow and readability of the section. While I kept the initial heading the same, some reformatting has greatly improved this section, especially as it provides a better setup for discussion of environmental discourse analysis – I appreciate your guidance here.

Film Synopses

I expanded the description clarifying the Lorax movie from earlier versions. An early draft of this paper had indeed discussed this more in depth, but I had cut it due to feelings of flow. However, upon further reflection, you are right that it is absolutely needed in order to maintain a clear distinction for readers less familiar with this then ourselves. Further focus was given to the section on Audrey and how she is different from other Thneedville residents.

For Tomorrow I added those additional citations as requested – It is important for all information presented to be verifiable, so I appreciate you pointing out where citations may be lacking.

Analysis of public reactions

You are correct about table 1 – I offered some more clarification on the comment example criteria shown there. For the graphs, I maintained the original to offer easy comparisons, but also split it up as you suggested, which greatly enhanced the readability of this section. You are correct that the location of the intersectionality information is awkward, but it is somewhat intertwined with this structure now, and finding a way to properly integrate this elsewhere is proving more difficult than expected.

For the skip gram I am considering modifications that would make it more readable, but this too is more difficult than expected, as I am not satisfied with the current accessibility of the image but am unsure how to improve it to be more accessible while still getting the data across.

Environmental Discourse Analysis

The order has been changed as suggested. I am trying to add more discussion of the movies in my revision per your suggestion, but I also wish to avoid repetition with the earlier synopses.

Environmental Catastrophe

This has been condensed and partially revised to increase clarity per your suggestion.

Environmental Storytelling

I attempted to be more specific without disrupting flow and keeping things streamlined.

Conclusion

I have tried to search for areas to bring these sources up earlier in more places than the start – However doing so without causing streamlining issues is a challenge.

Thank you for all your comments! They have made my revision a better paper, and I seek to fix and address what I so far have not yet been able to address.